# Animal Welfare and Meat Quality Assessment in Gas Stunning during Commercial Slaughter of Pigs Using Hypercapnic-Hypoxia (20% CO_2_ 2% O_2_) Compared to Acute Hypercapnia (90% CO_2_ in Air)

**DOI:** 10.3390/ani10122440

**Published:** 2020-12-20

**Authors:** Sophie Atkinson, Bo Algers, Joaquim Pallisera, Antonio Velarde, Pol Llonch

**Affiliations:** 1Department of Animal Environment and Health, SLU, P.O. Box 234, 53223 Skara, Sweden; kurrawong@hotmail.com (S.A.); bo.algers@slu.se (B.A.); 2Animal Welfare Program, IRTA, Veïnat de Síes, 17121 Girona, Spain; joaquim.pallisera@irta.cat (J.P.); antonio.velarde@irta.cat (A.V.); 3School of Veterinary Science, Universitat Autònoma de Barcelona, 08193 Cerdanyola del Vallès, Spain

**Keywords:** animal welfare, carbon dioxide, hypercapnia, hypercapnic-hypoxia, meat quality, nitrogen, pigs, slaughter, stunning

## Abstract

**Simple Summary:**

Animals must be stunned before slaughter to avoid fear, pain, and distress. In pigs, the most extensively used method is exposure to hypercapnia (high (>80%) concentrations of carbon dioxide (CO_2_)). However, it produces irritation of the mucosa and a sense of breathlessness, reducing the welfare before slaughter. We investigated whether using hypercapnic-hypoxia (20% CO_2_ and less than 2% O_2_) reduced aversion and discomfort compared to hypercapnia, and whether the quality of the stunning was adequate, meaning that no animals regain conscious after stunning. Moreover, we compared the impact of both stunning gases for meat and carcass quality. Our results suggest that both gases provoked aversion and discomfort, but these were lower in pigs stunned with the N_2_ mixture compared to high CO_2_. On the other hand, the stun quality of the N_2_ mixture was poorer than high CO_2_ stunning, given that more animals regained consciousness before sticking with the N_2_ gas mixture. The stunning quality of the N_2_ mixture, however, was improved when oxygen concentration was below 2%. Meat quality was slightly poorer in N_2_ stunning compared to high CO_2_, with a higher percentage of carcasses showing pale, soft, and exudative pork.

**Abstract:**

This study assessed aversion, stunning effectiveness, and product quality of nitrogen and carbon dioxide (CO_2_) mixtures used for stunning pigs. A total of 1852 slaughter pigs divided into two similar batches was assessed during routine slaughter in a Swedish commercial abattoir using either hypercapnic-hypoxia (20% CO_2_ and less than 2% O_2_; 20C2O) or hypercapnia (90% CO_2_; 90C) gas mixtures. Behavioral indicators of aversion and discomfort were recorded. After exposure, the stunning quality was assessed through brainstem reflexes. After slaughter, the pH and electric conductivity of carcasses were assessed to estimate the incidence of pale, soft, and exudative (PSE) pork, and the presence of ecchymosis were inspected. Compared to 90C, pigs exposed to 20C2O showed a later (*p* < 0.05) onset of behaviors indicative of aversion, and a lower (*p* < 0.01) incidence of breathlessness. However, unconsciousness (i.e., losing posture) appeared earlier (*p* < 0.01) in 90C compared to 20C2O. In 90C, all (100%) pigs were adequately stunned, whereas in 20C2O a 7.4% of pigs showed signs of poor stunning, especially when oxygen concentrations were >2% (*p* < 0.001). The percentage of PSE carcasses was higher (*p* < 0.01) in 20C2O than 90C. In conclusion, compared to 90C, 20C2O reduced aversion and discomfort but showed lower stun effectiveness, especially when O_2_ was above 2%, and a slightly poorer pork quality.

## 1. Introduction

Exposure of pigs to hypercapnia (high concentrations of carbon dioxide (CO_2_)) is a predominant method for stunning at commercial slaughter. Positive aspects of the system allow small groups of pigs to be efficiently moved towards the stunning unit using mechanical push gates, which minimizes the handling stress involved with human contact. Allowing pigs to remain in groups during preslaughter handling and stunning also respects the natural herd instincts in pigs to remain in social contact with one another, thereby minimizing fear and stress caused by isolation and close human contact, often associated with electrical stunning. Close human contact and restraint of pigs individually during stunning has been associated with causing preslaughter handling stress and meat quality defects, such as pale, soft, and exudative (PSE) and blood splash [1,2].

Despite the fact that group handling and mechanical driving systems minimize pig stress and optimize stun and production efficiency, exposure to hypercapnic gas as a stun method causes some welfare insults to pigs [3], and there is a feasible scope for research to further improve animal welfare [4]. The main animal welfare concern is that hypercapnia produces irritation of the nasal mucosal membranes, hyperventilation, and breathlessness (perceived as a sense of lack of air) [5,6]. Velarde [2] and Raj and Gregory [7] found that hypercapnic stunning leads to an elapsed time period where loss of consciousness occurs. Raj and Gregory [8], Velarde et al. [6], and Verhoeven et al. [9] all found that during this time period, induction of unconsciousness is often considered to be aversive and stressful, indicated by a series of observed behaviors. Verhoeven [9] found that sniffing, retreat attempts, lateral head movements, jumping, and gasping all occurred before loss of consciousness, which was confirmed via EEG latency, indicating ceased brain activity (when pigs were stunned in 80 or 95% CO_2_ atmospheres). In this study, the time to loss of consciousness varied from 47 ± 6 s to 33 ± 7 s in 80 and 95 CO_2_, respectively, during which the above-mentioned behaviors indicated that aversion to CO_2_ was highly probable. This is further supported by Nowak et al. [10], who found high lactate levels in pigs exposed to 80% CO_2_, indicating stress. This can be related to air hunger when demand for ventilation exceeds the capacity to provide it, causing chest tightness and respiratory distress, which is always an unpleasant experience and a serious concern for animal welfare [4,11].

In contrast to hypercapnia, pigs show less or no aversion during induction to unconsciousness when inhaling a hypoxic atmosphere saturated with inert gases, such as argon in a hypoxic atmosphere (less than 2% O_2_ by volume in atmospheric air) [12,13,14]. As argon has a higher density than air, it is possible to use it in current commercial gas stunning systems. However, due to its negligible presence in the atmosphere (<0.01%) the cost of this gas is prohibitive, and the practicality of using argon in commercial slaughterhouses is currently limited in comparison to CO_2_. In contrast, nitrogen (N_2_) (also an inert gas), is more readily available than argon due to its large presence in atmospheric air (79%). Unfortunately, N_2_ has a lower density than air, so the maximum concentration that can be confined in a commercial open pit is 85% in atmospheric air [15]. However, mixing N_2_ with CO_2_ improves the gas stability and uniformity. In addition, the mixture of these two gases (CO_2_ and N_2_) in different proportions leads to hypercapnic-hypoxia causing a quicker depressive effect of the central nervous system (CNS) [13] compared with hypoxia. Llonch et al. [16] tested aversion responses in pigs exposed to a range of mixtures during stunning containing 15 to 30% CO_2_ in a N_2_ saturated atmosphere. Pigs showed behaviors that indicated the less aversive concentration of hypercapnic-hypoxia contained 80% N_2_ and 20% CO_2_, which in that study as well as in other references from Llonch and colleagues was designated as 80N20C. However, in the present study, hypercapnic-hypoxia is referred to as 20C2O.

Meat and carcass quality of pigs stunned with N_2_ and CO_2_ mixtures has also been studied in experimental facilities. Llonch et al. [17] found pigs stunned with 20C2O had a lower pH than hypercapnia (90C) at 45 min after slaughter (i.e., no presence of pale, soft, and exudative pork (PSE)), but there was a higher presence of ecchymosis in the carcasses. Ecchymosis may occur due to the rupture of muscle capillaries induced by strong muscular excitation during induction to unconsciousness [1]. Llonch et al. [17] hypothesized that there might have been a relationship with the occurrence and severity of muscular movement and the fact that the pigs were exposed individually in a dip–lift stun system during the experimental conditions. A reduction of the severity of muscular activity is foreseeable if pigs can remain in groups during gas exposure by avoidance of isolation anxiety.

The aim of this study was to investigate the duration and range of behaviors that are indicative of pain, fear, and discomfort during stunning in a standard group stunning system under commercial conditions with an inhalation of 20C2O hypercapnic-hypoxia (20% CO_2_ with up to 2% O_2_ (20C2O) compared to hypercapnia (90% CO_2_ (90C)) and the effects on meat pH and conductivity and ecchymosis on the carcass.

## 2. Materials and Methods

The methods and design of this experiment complied with the EFSA Guidelines on the 12th June 2018 document no 10.2903/J.efsa.2018.5343, and they were approved by the Regional Ethics committee in Gothenburg on 2012-09-12 with the reference number 178-2012. Studies were conducted during routine slaughter in a commercial abattoir in Sweden with a slaughter capacity of 650 pigs/hour. Animal welfare and meat and carcass quality measurements were assessed in two batches of slaughterweight pigs (including males and females) with the same genetic background of the breed halothane negative “PigHam” strains (Hampshire sire lines with Landrace × Yorkshire sows). Pigs were from different farms of origin but were all submitted to the same handling once at the slaughterhouse. Pre-slaughter handling included unloading from the truck, lairage for 2 to 6 h in the pens, and handling to the stunning unit through an automatic push gate system without human intervention. Before slaughter, one batch of 1025 pigs was conventionally stunned with hypercapnia (90% CO_2_ (90C)) on one day, and the second batch of 827 pigs was stunned with hypercapnic-hypoxia (20% CO_2_ and 2% O_2_ (20C2O2O)) on another day. The average live-weight of pigs studied in both batches ranged from 90 to 95 kg (92.3 ± 1.92 kg), with no significant differences between batches. The stunning unit was a paternoster system (Butina, Holbaek, Denmark) contained in a deep cement walled pit which housed six metal crates connected in a gondola system that moved down and up through the gas gradient inside the pit (Figure 1).

### 2.1. Technical Aspects of Gas Filling the Pit

Pigs stunned with 90C were slaughtered on a different day than those with 20C2O. In an ideal situation, a cross-over study would be suitable, where pigs slaughtered each day are stunned with either 90C or 20C2O. However, considering the time required to fill the pit with 20C2O, it conflicted with the commercial requirements of the slaughterhouse, and the pit could not be filled twice in the same day with different gas mixtures.

In the 90C study group, a constant CO_2_ gradient during processing (at least 80% in the first stop and 90% at the bottom) was achieved by injecting CO_2_ gas through a permanently installed supply valve located at the bottom of the pit automated by built-in sensors (Figure 1). The sensors were positioned in the pit at a height corresponding to the top of the stun box when in the lowest position (i.e., approximately 1m from pit bottom) and at the first stop, which gave constant digitally displayed readings for CO_2_ concentrations on the machine. The gas was supplied by Linde Gas AB (Lidingö, Stockholm, Sweden) via a permanently constructed refillable supply tank.

During the second study group (20C2O), N_2_ was supplied into the pit to displace air and reduce O_2_ levels to below 2%. The target maximum concentration of O_2_ was set at 2% at the first stop (2.2 m down) and even lower at the bottom stop (5.6 m down) in the stunning unit. Gas was supplied through a built-in filling system based on 12 bottle clusters containing a mixture of 20% CO_2_ and 80% N_2_ (Praxair, Köping, Sweden). The gas bottles contained 155.6 m^3^ gas registered for food quality usage. A total of nine Mapcon^®^ ND20 bottles (Praxair Sverige AB, Köping, Sweden) were used during the trial at an initial pressure in the cylinder of 200 bars. The gas from the bottles was transferred into the Butina unit using one flexible metal pipe with an inner diameter of 25 mm with regulators (U13 F20 200/20 bar) operating at 3.5 bars and attached to a heater to warm the gas. From the heater, three pipes (two 12 mm and one 14 mm diameter) connected into the base of the pit delivering gas into the system. Two MAP Check 3 loggers (Dansensor, Ringsted, Denmark) were used for continuously measuring CO_2_, N_2_, and O_2_ concentrations inside the pit. The Map Check loggers were connected with pipes to sample and measure at the two stops (2.2 m and 5.6 m from the top of the pit), which measured six gas samples every minute (one every ten seconds). The supply of the 20C2O gas mixture was regulated manually according to the actual gas concentrations at both sampling points.

Additionally, four battery operated temperature loggers (Ibutton data logger, ThermoChron, IDC, Barcelona, Spain) were placed at 2.2 m (two Ibuttons) and 5.6 m (two Ibuttons) down in the pit. These loggers recorded temperature every 5 min and stored the information, which could be downloaded after the end of the trial.

### 2.2. Duration of the Gas Stunning Exposure

The time of exposure, considered from the time the box started moving down till the release of the stunned pigs into the conveyor belt, was different between the two gas mixtures. The time of exposure during 90C stunning ranged from 193 s to 259 s (237.3 ± 11.34 s), following the recommendations of the manufacturer (Butina), which is to keep pigs in the stunning unit for no less than 180 s. Exposure times were also influenced by the time it took to load each pig group, which varied from one group to another. In 20C2O 2stunning, exposure time ranged from 310 s to 453 s with an average time of 334.0 ± 23.56 s. This time was chosen based on the recommendations from Llonch et al. [18], who used the same gas mixture to stun pigs. In addition, a limited pilot study (6 groups of 4 pigs each) was performed at the beginning of the study, where pigs were exposed for 330 s to 20C2O with less than 2% of O_2_ in atmospheric air. This was to check that the exposure time was suitable in this system to ensure pigs did not show signs of recovery during the stun-to-stick interval nor during bleeding. Exposure time was recorded per filmed group from recording the time each box was seen moving through the gas gradient.

### 2.3. Behavior Recording during Gas Stunning

A battery-powered video recording system was installed in two of the six (one every three) stunning crates. Two waterproof surveillance cameras with infra-red capacity were used (Zavio Inc, Hsinchu City 300, Taiwan). Each camera was powered via four 12 V waterproof batteries attached in parallel, rewired from the original 230 V AC system. The batteries were housed in custom-made metal boxes bolted onto the top of the roof of the crates. The camera domes were bolted in the upper corner inside the crate, aimed to film towards the opposite corner. Image testing was performed during a pilot study to confirm that both provided good quality for pig behavior analysis. After the end of the trial, image data from both batches (90C and 20C2O stunning) was downloaded for posterior analysis.

### 2.4. Stun Quality Assessments

Two people continually (every 10 s) monitored the stun quality of all pigs after exiting the stunning unit and during shackling, hoisting, and sticking until 60 s after sticking without any sign of recovery. Group sizes in the stun-boxes were recorded by counting the number of pigs in each batch delivered from the stun-crate. The stun-to-stick interval (timed with a stopwatch) was considered as the interval in seconds between when the crate stopped, just before the gate opened for delivery of stunned pigs, until sticking. This was timed sequentially for each pig in the group. All pigs were chest stuck (where all major blood vessels in the thorax were severed) as of routine procedure in slaughter plants with one of six blood collecting knives (Rotary Stick, Butina Anitec^®^, Holbaek, Denmark), which are stuck into the chest to collect blood until cessation of bleeding.

Criteria for stun quality developed by Atkinson et al. [19] were used to score the risk level for recovery and animal welfare concern for each pig, ranging from 0 (no risk) to 4 (high risk) as indicated in Table 1. If pigs were in a state of whole-body relaxation, and there was no evidence of rhythmic breathing, righting reflex, or eye responses to stimulation, the stun was considered as correct and not at risk for poor animal welfare (score 0). Pigs that showed symptoms scored 1 or above were closely examined every 10 s until sticking was completed to ascertain if inadequate stunning might occur. Pigs with a stun level of 2 or 3 were not considered to be conscious, but the signs indicated risk for recovery and a justification for re-stunning. Pigs graded as level 4 indicated probable consciousness and a high concern for animal welfare and were immediately re-stunned. Re-stunning was performed with a captive bolt weapon shot to the forehead (brand type Accles and Shelvoke, London, UK, Cash Magnum 0.22 caliber).

### 2.5. Analysis of Filmed Behavior during Stunning

From the 1025 pigs stunned with 90C, 115 groups (393 pigs) were filmed for behavioral analysis. In hypercapnic-hypoxia (20C2O) stunning, from the 827 pigs stunned, 76 groups (268 pigs) were filmed for behavioral analysis.

A list of behaviors with the help of previously described definitions from Llonch et al. [16] was developed and prepared in an ethogram (Table 2). The film footage allowed behaviors to be viewed and recorded in timed sequence of occurrence, from when pigs entered the stunning unit until they were released. Behavior observation was performed individually for each pig, the individual pig being the experimental unit, so each video footage was observed at least the number of pigs in the gondola. The same pig was continuously observed throughout the gas exposure until the group of pigs was released. Unconsciousness was estimated to occur once posture was lost with head relaxation. A time period where pigs potentially could feel pain and fear (defined as a discomfort period), evidenced by aversive behavior, was considered to occur from when the first pig showed escape/retreat behaviors (through erratic turning, pushing, and moving about the stun box) until when the last pig was lying prone with its head in a relaxed state.

Gas exposure times were calculated by watching the time stamped films and recording the time from when the crate started descending and moving through the gas gradient to when pigs were released from the crate.

### 2.6. Assessment of Meat and Carcass Quality

After bleeding, carcasses were moved to the clean zone of the slaughterhouse to be processed. Each carcass was scalded and eviscerated and, after splitting, stored in chilling rooms at approximately 2–3 °C room temperature.

Meat quality measurements were assessed in 183 carcasses from pigs stunned with 90C and 223 carcasses from pigs stunned with 20C2O. At the end of the processing line, and before entering the chilling room, two meat quality technicians recorded carcass and meat quality parameters. Meat quality was assessed on the loin muscle (longissimus thoracis (LT)), at the level of the last rib, on the left side of the carcass. Muscle pH at 45 min post mortem (pH45) was assessed. Twenty-four hours later, pH (pHu) and electrical conductivity (EC) were assessed before carcasses were issued from the chilling room. The muscle pH45 and pHu were measured with a portable pH meter (Knick, Berlin, Germany) equipped with a Xerolyt electrode. EC was measured at the last rib level using a pork quality meter (PQM-I, INTEK, Aichach, Germany). LD muscles showing pH45 < 6.00 and ECu > 6.00 were classified as pale, soft, and exudative (PSE) meat, whereas carcasses presenting pHu > 6.00 were classified as dark, firm, and dry (DFD) [20]. Carcass quality was assessed by visually inspecting the presence of ecchymosis, defined as blood areas greater than 1 cm and darker in color [21], located in the caudal third of the hams. The whole ham could not be assessed because the slaughter company did not want to remove all the skin of the ham (only one third), fearing that it would impact the quality of the product.

### 2.7. Statistical Analysis

Data were analyzed with the Statistical Analysis System (SAS 9.3, SAS Institute Inc., Cary, NC, USA, 2002 to 2010). Latency measures (time of the first retreat attempt, escape attempt, gasping, loss of posture, muscle excitation, and laying), as well as meat quality parametric measures (pH45, pHu, and EC) were checked for normality (PROC UNIVARIATE) and analyzed with a mixed model (PROC MIXED) with the “gas mixture” as a fixed effect and the “group” as a random effect. Additionally, the “camera” was included as a random effect in behavior models, whereas in meat quality models, the “animal” was considered a random effect. The binomial data with non-parametric distribution was analyzed using a generalized linear model (PROC GENMOD) using a binomial distribution. The variables assessed using this model were percentage of animals showing retreat attempts, escape attempts, gasping, and muscular excitation during exposure; percentage of animals showing stun quality level from 2 to 4; and percentage of carcasses showing PSE meat and presence of ecchymosis. The “gas mixture” was considered as a fixed effect in all models. Oxygen concentration (either above or below 2% in atmospheric air) and the “stun-to-stick interval” were considered as covariables in the stun quality model. In all tests, level of significance was set up at *p* < 0.05.

## 3. Results

### 3.1. Gas and Temperature Recordings

During 90C stunning, CO_2_ concentrations remained at 81.6 ± 0.76% at the first stop (2.2 m down) and 92.2 ± 1.32% at the bottom (5.6 m down), and the O_2_ concentration was always under 2% in both sampling points. During 20C2O stunning, N_2_, CO_2_, and O_2_ concentrations were 81.0 ± 0.34%, 17.3 ± 0.44%, and 1.29 ± 0.33% at the first stop and 81.3 ± 0.37%, 17.4 ± 0.41%, and 1.3 ± 0.28 at the second stop, respectively. However, gas concentrations fluctuated at 2.2 m and 5.6 m sampling points, as shown in Figure 2.

The temperature inside the pit during both slaughter batches (90C and 20C2O) ranged between 14.6 and 18.5 °C at the first stop (2.2 m), and 13 and 16 °C at the bottom (5.6 m), without significant differences between batches.

### 3.2. Behavior Analysis during Gas Mixture Exposure

During 90C stunning, it was possible to analyze all the recorded groups (*n* = 115) for onset, duration, and cessation of behaviors. Although individual pigs within a group showed some degree of variation in duration and intensity, all pigs did display behaviors listed in Table 2. In total, 61 groups were filmed with four pigs, 48 with three, and six with 2.

The most prominent behaviors included raising the head and snout upwards towards the cage roof. Even when pigs lay on the side, the head was often lifted upward towards the cage roof, and the mouth was wide open. This was followed by a series of struggling behaviors including erratic jumping up and down, body jerks, thrusting against cage walls, and then lurching forwards and/or jumping upwards and downwards with aggressive body movements. This was ensued by a loss of posture followed by further muscle excitations while laying, then eventually by frequent then infrequent episodes of gagging. Table 3 shows the sequential occurrence of behaviors from the stun box entrance until loss of posture and thereafter.

In 20C2 stunning, from the 76 groups filmed, only 46 provided images of sufficient quality for a behavior analysis during exposure. The 30 other video recordings had to be discarded due to insufficient quality because the camera was either dirty (dirt splashing over the camera lens) or loss of focus when hit by a moving pig. Number of pigs per group varied from one to four pigs. The number of groups with respective group size was as follows: 43 groups with four pigs (57%), 24 groups with three pigs (33%), 1 group with two pigs (1%), and 7 groups with one pig (9%). Although individual pigs within a group showed various degrees of duration and intensity of the behaviors listed in Table 2, all pigs showed exploration, retreat/escape attempts, struggle, and gasping sequentially before the onset of muscle excitation while standing.

During both gas stunning treatments, all pigs observed showed escape and retreat attempts soon after starting the descent into the pit. The percentage of animals performing retreat or escape attempts, loss of posture, laying, and muscle excitation was not different between both gas mixtures. However, gasping occurred more frequently (*p* < 0.001) in pigs exposed to 90C (92.9%) compared to 20C2O (61.7%). Muscular excitation was more frequent (*p* = 0.017) in 90C (99.1%) stunned pigs compared to 20C2O (91.4%). As shown in Table 3, the average time for pigs to show the first retreat or escape attempt, loss of posture, gasping, muscle excitation, and laying was shorter in pigs exposed to 90C compared to 20C2O (*p* < 0.001).

### 3.3. Stun Quality

In 90C stunning, 500 pigs were assessed for stun quality and 300 pigs for stun-to-stick interval, whereas in 20C2O, 825 pigs were assessed for both stun quality and stun-to-stick interval. The average stun-to-stick interval for both 90C and 20C2O was not significantly different and together was 62.6 ± 0.59 s (ranging between 16 and 172 s), a similar range for both gas mixtures differing only according to the pig order of sticking after stunning (*p* > 0.05). In this regard, the average stun-to-stick interval for pigs stuck first, second, third, and fourth was 46.5 ± 0.66 s, 59.6 ± 0.89 s, 70.7 ± 0.68 s, and 81.0 ± 0.88 s, respectively, which did not differ (*p* > 0.05) according to stunning treatments (90C vs. 20C2O).

After 90C stunning, all pigs (100%) were scored as adequately stunned (stun quality level of zero or one), as no pigs showed gasping, muscular excitation, corneal reflex, and rhythmic breathing. In 20C2O stunning, 737 pigs (92.6%) were considered adequately stunned (stun level zero or one), and 59 (7.4%) showed some sign of inadequate stunning (stun level two or three and four), and they had to be re-stunned. This was associated with O_2_ levels during 20C2O exposure, so the percentage of inadequately stunned pigs when atmospheric O_2_ concentration was above 2% was higher compared to pigs exposed to an atmospheric O_2_ concentration lower than 2% (*p* < 0.001). From those inadequately stunned (level two, three, or four), 21 pigs showed gasping or kicking (scored a stun level of two); 41 pigs were scored three because they showed either corneal reflex or rhythmic breathing; and one pig was scored four because it showed corneal and righting reflex. The total number of pigs that showed each sign of poor stun quality is detailed in Table 4. The stun-to-stick time had no significant effect (*p* > 0.05) over stun quality scoring in any of the two gas mixtures.

### 3.4. Meat and Carcass Quality

As shown in Table 5, the pH45 of the loins was lower in animals stunned with 20C2O compared to 90C (*p* < 0.001). Conversely, the pHu was not different (*p* > 0.05) between 90C and 20C2O. The average EC was higher (*p* < 0.001) in 20C2O compared to 90C. According to Table 6, the percentage of carcasses categorized as PSE was higher (*p* = 0.002) in 20C2O compared to 90C. No carcasses were categorized as DFD. No carcasses from pigs stunned with 90C showed ecchymosis, whereas one carcass of the 20C2O stunning group (*n* = 223) showed it.

## 4. Discussion

Stunning before slaughter is performed to reduce fear in the animal, as well as to prevent pain and suffering during exsanguination. According to EFSA [6], stunning must induce immediate and unequivocal loss of consciousness and sensibility or, if this is not immediate, the induction should not cause fear, pain, or suffering in conscious animals. In addition, unconsciousness must last until brain death is achieved due to bleeding. For the stun method to be acceptable, it should also ensure a satisfactory level of carcass and meat quality. In this study, we examined two gas mixtures for stunning pigs (90C and 20C2O) to compare (1) the effects on aversion and discomfort during the induction to unconsciousness, (2) the capacity to maintain unconsciousness after stunning (stun quality), and (3) the meat and carcass quality.

### 4.1. Induction to Unconsciousness

In the behaviors analyzed during both gas stunning batches, pigs were not only aversive to both gas mixtures (90C and 20C2O) but also showed behavioral indications of discomfort (pain and distress) in nearly all groups (albeit at varying periods). This was more evident in pigs stunned in 90C compared to 20C2O. Pigs frequently started lifting the head upwards, pointing the snout towards the cage ceiling. They also often attempted to escape from the cage, with the head stretched up and mouth wide open, while struggling to avoid falling over.

Such behavioral signs of discomfort should be considered a welfare concern. According to animal welfare legislation [22], animal fear, pain, and suffering should be avoided during stunning and killing, highlighting the urgency to seek alternatives to CO_2_ stunning/killing [23].

The first sequence of behaviors shown during gas exposure in both stunning methods were retreat and escape attempts, both considered signs of aversion [8,24]. The first appearance of aversive behavior could be due to the inhalation of the gas mixture. However, the displayed aversion in hypercapnic-hypoxia stunning may not be due to acidification of the nasal cavity, as would happen with hypercapnia, as the threshold of CO_2_ concentration has been found to be between 20 and 30%, and severe hyperventilation would have been seen [25], which was not confirmed. It is also possible that the environment in the gondola, including new sounds and smells, could provoke a first aversive reaction. For instance, the sound of the chains against the cogs came suddenly when the gondolas moved through the machine, which could be heard via the digital sound recorders placed in the crates. Additionally, the sound of the gas injection was sudden and intermittent, and sometimes vocalizations from pigs from below gondolas could be heard when pigs were in the upper level. Given the conditions of this study, it was not possible to determine the drivers of escape and retreat attempts, and whether aversion was provoked by inhalation of a small concentration of the gas mixture or by the system environment, or both. In any case, from our results, it can be confirmed that the prevalence of animals showing these signs of aversion were not different between the two gas mixtures, but they appeared earlier in 90C than 20C2O. According to this, it is plausible that the onset of aversive response has to do with the level of CO_2_ concentration, being quicker when the CO_2_ concentration is higher. These results are in accordance with those from Velarde et al. [2], Llonch et al. [16], and Verhoeven et al. [9], who found that higher concentrations of CO_2_ (e.g., from 70% to 90%) trigger a quicker aversive response than mixtures containing lower concentrations of CO_2_ (from 30% to 80%).

Gasping is an indicator of breathlessness during inhalation of gas containing CO_2_ [2]. According to our results, pigs exposed to 90C showed a higher prevalence and a quicker onset of gasping compared to 20C2O. Llonch et al. [17] also found that gasping occurred more frequently and appeared earlier in animals exposed to 90C compared to 20C2O, the difference of which was even higher (87% vs. 19%) compared to results from this study (93% vs. 62%). This effect of CO_2_ concentration on gasping was already stated by Gregory [26], suggesting that the residual medullary activity in the brainstem that induces gasping when the atmospheric concentration becomes hypercapnic increases with higher concentrations of CO_2_.

The time to lose posture is considered the first behavioral indicator of the onset of unconsciousness [8]. In our study, loss of posture was the time when animals showed the inability to stand in an upright position, but it did not necessarily mean that they started to lay, as in numerous occasions pigs struggled to maintain their position. Therefore, we also considered the time in which pigs started to lay when the whole body was resting and the head was on the ground and not flexing upwards. Considering either the onset of loss of posture or laying behavior, pigs exposed to 90C showed a quicker response than 20C2O, suggesting that higher concentrations of CO_2_ induce unconsciousness faster, confirming the results of similar studies conducted in experimental conditions [16,17]. However, in this occasion, the rapidness to induce the loss of posture (even considering laying behavior) was quicker compared to [16,17]. Our hypothesis is that in the present study conducted in commercial conditions, the circular movement of the crates never stopped, which helped improve the gas uniformity within the pit (upper vs. lower depth). In the studies of Llonch et al. [16,17], pigs were stunned with a dip–lift system with a single cage moving up and down, which may not have contributed as much to uniformity within the pit. Conversely, in the commercial paternoster system, the gas concentration was more homogeneous within the pit, which makes it that pigs probably met the highest CO_2_ and the lowest O_2_ concentration in a shorter time after starting the exposure compared to the dip–lift system, facilitating a faster effect in the paternoster system compared to the dip–lift.

Duration of muscular excitation was calculated as the time between when the first pig started to move uncontrollably until cessation of muscle excitation in the last pig of the crate. In other studies, the duration of muscular excitation is measured individually, but this was not possible in our study, as video recording did not allow the individual identification of pigs throughout the exposure. Still, results are in accordance with other studies [17,18] where muscular excitation lasted longer in pigs exposed to 20C2O compared to 90C. Raj [14] states that the acidification of the central nervous system, when inhalation of high concentrations of CO_2_, inhibits muscular excitation during induction. Therefore, the higher the concentration of CO_2_, the shorter the duration of muscular excitation.

There has been some debate on whether muscular excitation occurs as a voluntary response to the gas mixture, and is therefore indicative of discomfort, or if it is a period of involuntary movements provoked by a lack of modulation of the neuronal structures that regulates motor activity [27]. According to a report issued by EFSA [6], muscular excitation during CO_2_ exposure might be an aversive response, with implicit consciousness. In studies assessing the brain activity during exposure from 80% to 95% CO_2_ [9,28] and N_2_ and CO_2_ gas mixtures [18], evidence suggests that the start of muscle excitation occurs before significant changes in brain function appear, which could indicate that pigs were conscious. Furthermore, some of these studies [9,18] show evidence that most of the muscular excitation period occurs while brain activity is still high, indicating that the pig is still likely to be conscious.

### 4.2. Stun Quality

All (100%) pigs stunned with 90C were appropriately stunned after exposure and remained unconsciousness until brain death after sticking. According to instructions from the manufacturer (Butina), the minimum exposure time should be not less than 3 min, and the CO_2_ concentrations should be no less than 80% at the first stop and more than 90% at the bottom of the pit. As CO_2_ concentration and time of exposure were always higher than indicated by the manufacturer, it is not surprising that all pigs exposed to 90C were appropriately stunned.

The stun quality was satisfactory (no pigs recovered consciousness) despite the fact that the stun-to-stick interval was often longer than one minute, which is the maximum time recommended by EFSA [6]. This may be explained because most of the pigs were probably dead when they were released from the stunning unit. This supports the findings from Llonch et al. [18], who stated that the majority of pigs were dead after a 3 min exposure to 90C in a dip-lift system. The death of pigs after exposure to gas stunning safeguards animal welfare, as the stun-to-stick interval becomes irrelevant. For example, Atkinson et al. [19] found that a 90% concentration of CO_2_ stunning in six Swedish slaughter plants consistently provided adequate stunning (and probably death) despite stun-to-stick times longer than EFSA recommendations.

Conversely, 20C2O could not achieve the stun quality of 90C stunning, with 7.4% of pigs showing some level of inadequate stunning, which is also over EFSA’s recommendations of no more than 5% of pigs having signs of inadequate stunning [6]. It is worth noting though that only five of the 796 pigs assessed had a stun level of four, indicating a high risk of poor animal welfare. There are several reasons that could explain this lower stun quality of 20C2O. The most notorious reason is the concentration of oxygen inside the pit. According to our results, a presence of oxygen above 2% in atmospheric air was associated with a higher percentage of signs of recovery before sticking. The facilities used in our study were designed to be used with high concentrations of CO_2_. However, they may not necessarily be suitable to contain a modified atmosphere containing 80% N_2_ and 20% CO_2_, (with less than 2% O_2_). Gas mixtures containing high concentrations of N_2_ are difficult to contain because its lower molecular weight makes it difficult to contain in a pit [15]. In our study, there was a need for a constant supply of 20C2O, suggesting that the gas mixture was continuously escaping from the pit. Therefore, gas stunning facilities should be adapted to improve containment of N_2_ and CO_2_ gas mixtures where the O_2_ concentration can also be kept low. Our results suggest that if O_2_ levels can be kept under 2%, the likeliness of recovery after a 5.5 min exposure to 20C2O is dramatically reduced. The stunning quality should be further investigated with maintained low (<2%) oxygen levels before hypercapnic-hypoxia can be considered as a real alternative to hypercapnia

The time of exposure to the gas mixture has also an effect on the stun quality. Llonch et al. [18] recommended that for a good stunning quality using 20C2O, pigs need to be exposed to the gas for at least five minutes. The previous statement was based on pigs stunned with an individual dip–lift system without stops in an 8m^3^ pit volume. In the present study, similar to many other commercial abattoirs, the stunning unit encompassed six cages each loaded with a nominal group of 3 to 4 pigs, rotating in a 63 m^3^ pit volume and stopping at different points (and gas concentrations) inside the pit. A proof of that is the difference in O_2_ concentration between the first and second stop (2.2 and 5.6 m depth) shown in this study. Although we already extended the exposure time (by 30 s) from what was recommended by Llonch et al. [17], we still found some pigs with inadequate stunning. In order to reduce the likeliness of recovery, a further extension could be implemented, seeking for complete breathing cessation and death of pigs, as occurred in 90C stunning. In this sense, Raj [14] recommended an exposure time of at least seven minutes in pigs exposed to a mixture of inert gas (argon) and CO_2_. This extension may be controversial if implemented in commercial abattoirs, as it may decrease the speed of the processing line and limit the production capacity of the plant. However, this can be compensated by increasing the capacity of the gondola system, which can be achieved by increasing the capacity of the gondola (more pigs per cage) or adding more cages.

According to our results, no association was found between stun-to-stick interval and the stun quality under the conditions of this trial. However, it is also true that if pigs are not killed by the exposure to the gas, the likeliness of regaining conscious increases with longer periods between the end of stunning and exsanguination [6]. In both stunning methods, the stun-to-stick time for the last pig in the group exceeded the Swedish recommendation of 60 s (average was 1 min 21 s for 90C and 1 min 23 s for 20C2O). As 38 pigs from 20C2O showed gagging after exiting the stunning unit, the combination of pigs coming out of the stunning unit alive with extended stun to stick times, and the physical action of gagging (deep breaths) would have facilitated O_2_ inhalation and the associated recovery. Evidence that consciousness was regained after the stun-to-stick interval was that corneal reflex appeared just before sticking and not when exiting the stunning unit. As a corrective measure, some authors [14,18] have stated that in order to guarantee unconsciousness of pigs until brain death when exposed to N_2_/CO_2_ gas mixtures, additional killing methods are required, such as electrically mediated cardiac arrest.

### 4.3. Meat and Carcass Quality

The results obtained in this study are similar to research obtained in experimental trials, where conditions and study design were controlled. Llonch et al. [17] found a lower pH45 in carcasses from pigs stunned with 20C2O compared to 90C (6.66 ± 0.05 vs. 6.26 ± 0.06; *p* < 0.001). In that study, differences in pH45 were clearly due to the effect of a different stunning gas mixture. It was suggested that the longer muscular excitation period during the exposure to the gas mixture increased abruptly the energy demands of the muscular tissue, which is associated with the anaerobic metabolism of glucose into lactic acid. The accumulation of lactic acid into the muscle results in a more pronounced decrease of pH compared to pigs with lower energy requirements closely before slaughter [29,30]. The extended muscular excitation period of pigs stunned with 20C2O compared to 90C shown in our study confirms that the significant pH reduction would also be due to the greater energy expenditure, likely due to a longer muscular activity before killing.

The decline of the muscular pH after slaughter is determinant for pork quality. The acidification of the muscular milieu provokes protein denaturalization, which leads the transformation of muscle into meat [29]. However, if the denaturalization process occurs too quickly, the capacity to hold water from proteins decreases, leading to an exudative meat [31]. PSE meat can be determined using different criteria. Two criteria that are frequently used are pH45 (lower than 6) and EC (higher than 6 µS at 24 h after slaughter) [21]. According to these criteria, 11 of the 223 carcasses (4.9%) from pigs stunned with 20C2O showed PSE meat, whereas no carcasses from 90C did so. Although the differences in percentage of PSE carcasses are significant, the number of affected carcasses is much lower than those reported by Llonch et al. [17] in experimental conditions (13% of carcasses with PSE). However, as the genetic background (and many other affecting variables) of both studies were different, it is difficult to compare results. PSE affecting nearly 5% of the carcasses from pigs stunned with 20C2O is still noteworthy, and the use of alternatives to 90C, as such as 20C2O, in commercial conditions should be promoted as long as product quality insults are addressed.

The presence of ecchymosis was used to evaluate carcass quality [21]. In the experiment from Llonch et al. [17], one out of every four carcasses from pigs stunned with hypercapnic-hypoxia showed ecchymosis in the hams. According to our results, no carcass from 90C stunned pigs showed ecchymosis, and it was insignificant in 20C2O, as only one out of the 223 carcasses had ecchymosis. These results indicate that carcass quality is not affected when nitrogen and CO_2_ gas mixture stunning is performed under commercial slaughter. However, these results should be taken with care because the hams were not assessed completely, being that two thirds of the ham were covered by skin and could not be assessed. Thus, further studies would be needed to confirm these results.

## 5. Conclusions

Pigs exposed to both 90C and 20C2O did exhibit behaviors indicative of discomfort. The sequence of observed behaviors was similar between both individuals and groups of pigs stunned in both gas mixes. Pigs exposed to 20C2O showed a reduced aversion response and a lower sense of breathlessness compared to 90C gas stunning. However, the time to reach behavioral indicators of unconsciousness, which entails a potential for discomfort, took longer in 20C2O compared to 90C. The stun quality, assessed by the brain stem reflexes, resulting from an average 5 min 30 sec exposure to 20C2O, was slightly poorer compared to a four min exposure to 90C. However, as signs of recovery in hypercapnic anoxia stunning are greatly dependent on gas exposure time and O_2_ concentration, an extension of the time of exposure and a contention of O_2_ levels below 2% may notably improve the stun quality. Stunning pigs using a 20C2O gas mixture under commercial conditions provoked a faster decrease of pH, increasing the incidence of PSE meat compared to 90C. However, the increase of PSE with 20C2O compared to 90C was lower than in previous studies, affecting less than 5% of the carcasses. Under the conditions of this study, no significant effects of hypercapnic-hypoxia on the presence of ecchymosis in the carcass were seen, but further experiments should be made in order to confirm these results. The results obtained in this particular abattoir suggest that 20C2O reduces aversion, but the gas exposure time must be extended and the O_2_ level kept below 2% all the time to reduce the percentage of animal recovery to an ideal zero percent.

## Figures and Tables

**Figure 1 animals-10-02440-f001:**
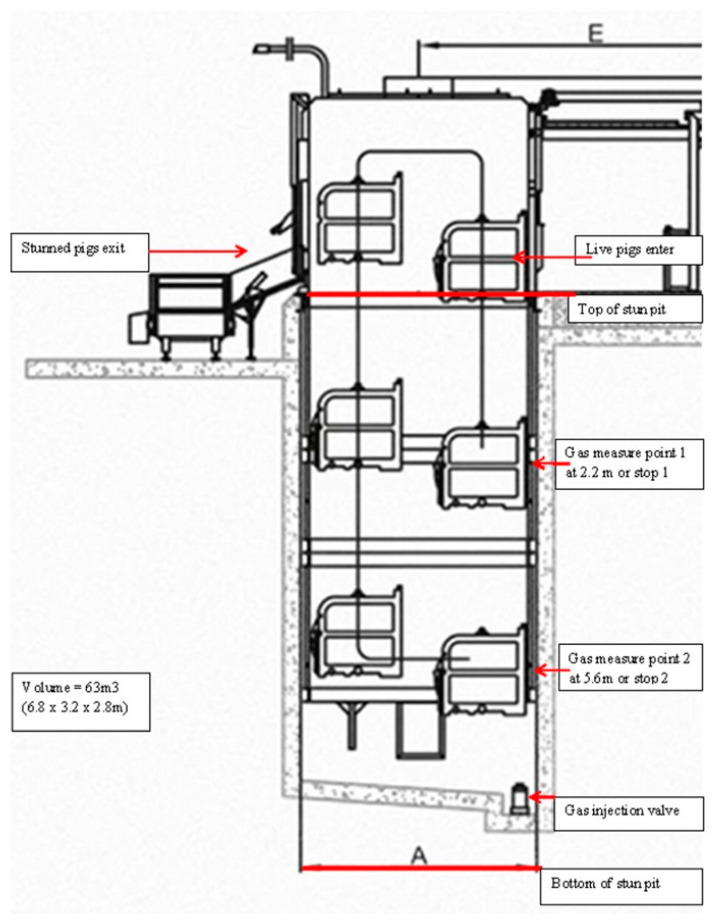
Detail of the stunning unit that describes the location of the gas sensors and the gas supply valve.

**Figure 2 animals-10-02440-f002:**
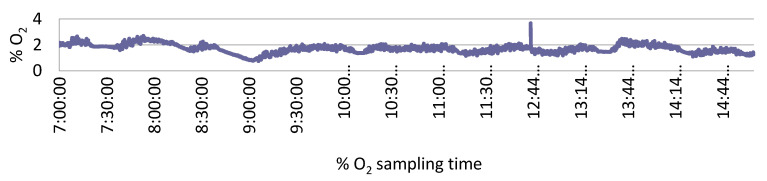
Oxygen (O_2_), nitrogen (N_2_), and carbon dioxide (CO_2_) concentrations at the bottom sampling point (5.6 m down), during the 20C2O gas stunning batch. The spike observed at 12:44 min corresponds to variation of gas concentration during the lunch break. After the break, slaughter did not start again until gas concentrations were restored.

**Table 1 animals-10-02440-t001:** Level of stun quality used and decision on captive bolt re-stunning after the end of gas stunning according to criteria used to monitor unconsciousness (extracted from Atkinson et al. [19].

Stun Quality Level ^1^	Description of Signs
0	If a pig showed no reflexes or signs mentioned below, it was considered as being in a state of deep unconsciousness and posed no risk for poor animal welfare
1	If a pig kicked, convulsed, or gasped infrequently (not more than twice before sticking) but showed no eye or pain reflexes when checked, it was considered adequately stunned but justified continual monitoring until sticking
2	If a pig displayed frequent (more than twice) gasps (opening and closing of the mouth with or without stretching of neck), kicks, or body convulsions but was found to have no eye or pain reflexes, it was re-stunned as a precaution to avoid recovery. The stun depth was considered as unacceptable due to the risk that the animal could recover
3	If a pig showed corneal or cilia blink reflex at sticking, with or without kicking or convulsions, it was immediately re-stunned, and the recovery risk was thought to be imminent, and the stun was considered inadequate
4	If a pig showed spontaneous blinking, righting reflex, vocalization, or pain reflex, it was considered as indicating some form of consciousness and a high risk for poor welfare, and the stun was considered inadequate

^1^ From level 1 to 4, the concern for poor animal welfare increases, while stun quality level decreases. Eye reflex includes either corneal reflex (by touching the cornea with a blunt object) or spontaneous blinking or both.

**Table 2 animals-10-02440-t002:** List of defined behaviors that were timed and registered after viewing recorded films of pigs stunned with hypercapnia (90C) or hypercapnic-hypoxia (20C2O).

Registered and Timed Onset of Behaviors Observed in the Group	Definition
1. **Exploratory**	Smelling floor, walls, or roof of pen and looking about the cage
2. **Retreat/escape attempts**	Backing, turning, or pushing into one another as they move around the cage in appeared attempt to find an exit
3. **Struggle**	Erratic jumping up and down, body jerks, thrusting against cage walls
4. **Gasping**	The hyperventilatory response to increased blood PCO2 characterized by maximal tidal volume and increase frequency of breathing. Usually very deep breath through a wide-open mouth, which may involve stretching of the neck.
5. **Fall**	Pig falls over with whole body because of the inability of the animal to remain in a standing position and considered an indicator of onset of unconsciousness
6. **Laying**	Pig is prone and head is relaxed where snout is not stretched or flexed upwards
7. **Muscle excitations**	Convulsive, spasmodic stretching and contracting movement of body, limbs or head after loss of posture
8. **Gagging**	Mouth opens and close periodically and occasional emitting of sounds similar to snoring while laying. It has been considered an indicator of deep unconsciousness [6]

**Table 3 animals-10-02440-t003:** Average time (s) of each behavior to appear for the first time during both 90C and 20C2O exposure.

Variable	90C			20C2O			*p*-Value
	*n*	Mean	SEM	*n*	Mean	SEM	
Retreat/Escape	108	6.26	0.250	42	8.24	1.062	0.0118
Lose posture	107	11.04	0.419	48	15.31	1.120	<0.001
Gasping	105	12.92	0.559	29	21.00	2.176	<0.001
Start muscle excitation	112	14.38	0.447	43	19.67	1.278	0.0049
Lying	112	18.16	0.490	43	25.37	1.429	0.015
Duration muscle excitation	111	141.55	2.711	43	249.74	9.507	<0.001

**Table 4 animals-10-02440-t004:** Percentage and number of pigs that showed gasping, corneal reflex, rhythmic breathing, and overall inadequate stunning (scored as stun level two, three, or four) according to the level of O_2_ during exposure to 20C2O.

Variable	O_2_ Above 2%	O_2_ Below 2%	*p*-Value
Percentage of gasping % (*n*)	12.0% (18)	3.3% (21)	<0.001
Percentage of corneal reflex % (*n*)	16.7% (25)	2.9% (19)	<0.001
Percentage of rhythmic breathing % (*n*)	10.0% (15)	3.1% (20)	<0.001
Percentage of righting reflex % (*n*)	0.7% (1)	0% (0)	-
Percentage of inadequate stunning % (*n*)	19.3% (29)	4.8% (31 pigs)	<0.001

**Table 5 animals-10-02440-t005:** Descriptive statistics (mean ± standard error) of the meat quality measures according to the gas used for stunning.

Meat Quality Measures		90C			20C2O		*p*-Value
	*n*	Mean	SEM	*n*	Mean	SEM	
pH45		6.42	0.016		6.17	0.014	<0.001
pHu		5.56	0.008		5.55	0.007	*n*/S
EC (µS)		3.74	0.036		4.34	0.064	<0.001

**Table 6 animals-10-02440-t006:** Percentages (%) of carcasses showing ecchymosis and considered as pale, soft, and exudative (PSE) meat according to the gas used for stunning.

Carcass Quality Measures		90C		20C2O	*p*-Value
	*n*	Percentage	*n*	Percentage	
Percentage (%) of Ecchymosis	0	0	1	0.45	-
Percentage (%) of PSE	0	0	11	4.82	<0.01

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
