# Peer review of "Animal Welfare and Meat Quality Assessment in Gas Stunning during Commercial Slaughter of Pigs Using Hypercapnic-Hypoxia (20% CO2 2% O2) Compared to Acute Hypercapnia (90% CO2 in Air)"

_animals, 2020, doi:10.3390/ani10122440_

Round 1

Reviewer 1 Report

My comments are directed towards a better communication of the findings.  I find no fault with experimental process or general construction of the communication.

Author Response

Revision Note to manuscript animals-1030363

The comments of the editor and reviewers are given below, with the authors’ response (AR) marked in Italics. Line numbers mentioned by the reviewers refer to the original manuscript, whereas line numbers in the response refer to the revised manuscript unless stated otherwise.

Authors Response (AR): We have amended the manuscript to all suggestions made by reviewers and have detailed the responses to the comments below.

Comments:

This concise manuscript describes a study investigating the effects of two kinds of environmental enrichment on behaviour (primarily play - a positive indicator of welfare) and other welfare indicators in pre-weaned and newly-weaned piglets.  It will be of interest to readers of the journal.  The effects found are modest and perhaps over-stated in the Abstract - as detailed below. There are minor errors in written English throughout the manuscript which need to be corrected.

Authors Response (AR): The suggestions have been addressed below and the manuscript has been reread and corrected for English by a researcher in UK.

Reviewer 1: Replace the title “Animal welfare and meat quality assessment in gas stunning during commercial slaughter of pigs using 80% nitrogen and 20% carbon dioxide compared to 90% carbon dioxide” by “Animal welfare and meat quality assessment in gas stunning during commercial slaughter of pigs using hypercapnic-hypoxia (20% CO2 2% O2) compared to acute hypercapnia (90% CO2 in air)”

AR: The title was changed as requested by reviewer.

Reviewer 1: It is my strong opinion that your research projects would be better communicated by changing the way you describe inhaled gasses to the format of hypercapnic-hypoxia (20% CO2, 2% O2) compared to 90% CO2 in air (hypercapnia)

AR: Authors agree with this recommendation and the way the two gas mixtures are referred throughout the document were changed accordingly.

In line with this suggestion, abbreviation of hypercapnic anoxia treatment was changed to 20C2O, instead of 80N20C, understanding that the true effect is driven by the concentration of carbon dioxide and the low concentration of oxygen. Although we agree that the term hypercapnia and hypercapnic-hypoxia may better reflect the effect of both gas mixtures on the organism, we prefer to keep acronyms to facilitate the reading.

The change of acronyms (80N20C of previous references to 20C2O in the present paper) was mentioned in the text to highlight that both nomenclatures refer to the same gas mixture (L95).

Reviewer 1: Gasping is defined in this paper in Table 2 point 2.4 (Contrary to the dictionary definition of the term). Gasp, as most commonly used in contemporary English, contains the implication of a short duration, non-ventilatory chest movement displacing a small fraction of lung tidal volume (+ sound) and I suspect that is not what you are observing in these hypercapnic pigs.

Gasping – would retain the implication of very small tidal volume as in a recovery from near drowning (functional tidal lung volume greatly reduced). If this is what is observed than the use of the word is correct but the phrase “high frequency low volume respiratory efforts” should be included in the definition table.

AR: We prefer to keep gasping as the definition of the behaviour observed. We understand that our understanding of Gasping is compatible with the definition of Merriam-Webster dictionary. However, we acknowledge that the definition should include a reference to hyperventilatory response to increased blood PCO2 and so we included it in the table (Table 2).

Line 102: This is not a compelling statement if this is a university animal review committee I stand corrected. However; it appears the methods and design of this experiment also comply with The EFSA Guidelines 12 June 2018 document no 10.2903/J.efsa.2018.5343 (6) and the previous version which is available to all readers globally and is a international standard.

AR: The ethics statement was completely rewritten according to reviewer recommendations, now reading “The experimental protocol the methods and design of this experiment comply with the EFSA Guidelines on 12th June 2018 document no 10.2903/J.efsa.2018.5343 and they were was approved by the Regional Ethics committee in Gothenburg on 2012-09-12 regional ethics committee in Gothenburg with the reference number 178-2012.” (L113)

Line 129: the reviewer suggest to change the sentence to “During the second study group (80N20C), N2 was supplied into the pit to displace air and reduce oxygen levels to below 2%.

AR: The statement was amended accordingly.

Line 177: This is a rare piece of equipment and this step requires more detail for a global audience. It also provides an explanation as to why it too more than 60 seconds to stick 4 pigs which a single operator working with a hand knife could do in less than 20 seconds on a metal table. I have not worked in a plant that recovers hog blood so perhaps I am out of touch. I found this photo on the internet and I would include one like it to clarify this step in the process.

AR: We prefer not to show a picture of the rotary blood collecting knives as this may not correspond to the facilities where this experiment took place. However, we described better the purpose of the equipment to explain its functionality to the general reader (L 193).

L198: The reviewer suggest to amend the sentence as “In hypercapnic-hypoxic stunning, from the 827 pigs stunned,…”

AR: This was amended accordingly (L 215).

L213: The reviewer suggests to include the sentence “The hyperventilatory response to increased blood PCO2 characterize by maximal tidal volume and increase frequency of breathing. Usually through a wide-open mouth, which may involve stretching of the neck.”

AR: This was changed (L 230).

L506: This is a statement so misleading that it constitutes an error in communication. Nitrogen is an inert gas and has no relevance to stunning other than its ability to exclude oxygen. The focus on nitrogen in the discourse of humane stunning is to obfuscate the experience of the animal with some technical aspect of the delivery and type of gasses. In this paper, pigs are unconscious from hypercapnia or hypercapnic-hypoxia. Nitrogen has nothing to do with the physiology that is resulting in the pig being rendered unconscious or dead. If we agree to tighten up the use of language in the document, this criticism can be easily avoided and the communication of the project improved.

AR: We agree with reviewer and we replaced “N2 gas mixtures” by “hypercapnic-hypoxia” (L523).

General comments:

There is no mention in the description or the discussion that pigs should not be able to detect 20% CO2. These authors have dealt with this issue in previous papers and we generally agree that the detection threshold for CO2 by pigs is somewhere between 20 and 30%. Detection is facilitated by acidification of the mucosa of the turbinates. If these pigs displayed aversion in the 20% CO2 2% O2 situation it was not due to acidification of the nasal cavity. They would have responded with hyperventilation within 3-4 normal tidal volume breaths, 60-90s (my best guess). I encourage this idea to appear in the discussion of behavioural responses.

AR: This idea is now included in the discussion (L 377).

I would like to see a carcass-cooling time graph for this establishment drip cooler as a figure in the paper in the meat quality section. This is a visual display generated from a deep muscle thermometer usually in the ham it is dependent on pig weight and a common quality monitoring device. Every drip cooler cools pig carcasses at slightly different time rates under a standard days kill load.

AR: Unfortunately, the temperature of the hams could not be systematically monitored in this trial and we cannot show the graph of the temperature drop. However, we understand that the conditions of this trial were on a commercial slaughterhouse with 24h cooling as regular practice. We believe that this may not have affected the variables assessed in this study, or at least, in the same way in both treatments.

This material was previously reported in a non-peer reviewed report (7) and I recommend including that information somewhere either as a reference or as a note.

AR: The non-peer reviewed report that is mentioned by the reviewer provides different data compared to this manuscript, as it was dedicated to report the main results and activities undertaken to the funder of this study. Given that data was preliminary and might differ to data shown in this manuscript we prefer not to refer to it, in order to avoid confusion.

Reviewer 2 Report

Overall, the manuscript is good and the premise is interesting, but information about the how the behavioral data was collected should be described in more detail. Additionally, the literature is predominantly from 1977 to 2015, although there is significant work in this field that has occurred in the past 5 years that could help shape the author’s interpretation of their data. Of concern is that they are using interpretations of words, such as “aversion”, while referencing articles from pre-1990, when this work has changed significantly over the past decade. This may inappropriately skew their conclusions. Some specific comments below:

Line 41: After reading the manuscript, you did not really do any assessment of “aversion” persay. I think you can say that the 80N20C reduced signs of agitation or escape behaviors (though, it didn’t really do that – it really just delayed onset of those behaviors). Please be careful in your wording and describe what you observed.

Section 2.5: Please provide more detail on how you assigned behavior scores. I would assume you would have to watch and score each video repeatedly so that you could observe each animal and it's behaviors (max of 4) on a continuous scale. However, this is my assumption because there is insuffient detail about how each of the 393 and 268 pigs were assessed (or less, since quite a bit of video was lost in the 80N20C group). If this was not the case, what did you do? Scan every x seconds to record what behaviors were done? Document the first time that any of the pigs in the gondola engaged in the behavior? In other words, was the behavioral experimental unit the gondola (115 groups and 76 groups) or the individual animal (393 and 268 pigs)?

Line 205: Again, you are measuring potential distress, evidenced by behaviors 2-3 (retreat and struggle), not aversion. Aversion has a specific meaning and requires specific testing to assess. An observational study like this one is not designed to assess diversion. I recommend that you remain consistent with using the behaviors described in Table 2 when discussing behavioral responses.

Lines 252-253: Please provide the ranges and SE for the measurements taken during the 80N20C stunning. Although the graphs are interesting, knowledge of the actual numbers would be invaluable. Also, what was the spike that occurred at 12:44:00?

Table 5: The SEM is reported twice for 80N20C.

Table 6: How many of these had to be restunned? I ask because I would have severe welfare concerns about recommending a method that has a significant risk of requiring restunning, but then ultimately results in a carcass that may not be usable (ultimate waste of animal life).

Line 349: Instead of “aversive”, I would state that “pigs were not only distressed by exposure to both gas mixes…..”

Line 359: The referenced articles do not appear to be aversion studies (and they are from 1996 and 1977). If you are going to try to make the case that these behaviors correlate to aversion, please reference studies that actually assess these behaviors specific to an aversion response. Preferably from more recent literature, where there has been a lot of work in this area. Otherwise, please remove references to aversion and replace with distress throughout this entire paragraph.

Lines 404: See comment above for behavioral materials and methods. As there were only a maximum of 4 pigs per gondola, how did you score them without visually tracking each animal individually when doing the scoring? Please be clear on if the experimental unit was the gondola versus the individual animal.

Line 417: Brain activity itself does not indicate consciousness. It is well documented that EEG does not correlated to consciousness in humans, so this statement is misleading.

Line 440: In the materials and methods, please describe how you were able to correlate the O2 levels in the chamber to the stunning success of a particular gondola.

Lines 448-449: I think your results indicate that this needs more investigation, given all of the potential limitations that you have described here for the use of 80N20C. It may show promise, but ethically, I have a hard time believing that this is a welfare refinement if 7% of the animals have to be restunned (welfare issue) and the use of the carcasses are questionable (waste of life) over a well-characterized technique that had no animals that required restunning and no rejection of carcasses.

Line 517: How did you define “breathlessness” (and aversion)? There are too many terms coming in during the results and discussions which have not been defined (or measured) in the study design. If you are using them to interpret the data, please describe which behaviors you are using to correlate to these terms. Ideally, you will continue to use the objective terms that you have defined in the materials and methods, instead of subjective interpretations of these behaviors.

Author Response

Revision Note to manuscript animals-1030363

The comments of the editor and reviewers are given below, with the authors’ response (AR) marked in Italics. Line numbers mentioned by the reviewers refer to the original manuscript, whereas line numbers in the response refer to the revised manuscript unless stated otherwise.

Authors Response (AR): We have amended the manuscript to all suggestions made by reviewers and have detailed the responses to the comments below.

Reviewer 2: Overall, the manuscript is good and the premise is interesting, but information about the how the behavioral data was collected should be described in more detail. Additionally, the literature is predominantly from 1977 to 2015, although there is significant work in this field that has occurred in the past 5 years that could help shape the author’s interpretation of their data. Of concern is that they are using interpretations of words, such as “aversion”, while referencing articles from pre-1990, when this work has changed significantly over the past decade. This may inappropriately skew their conclusions. Some specific comments below:

Line 41: After reading the manuscript, you did not really do any assessment of “aversion” persay. I think you can say that the 80N20C reduced signs of agitation or escape behaviors (though, it didn’t really do that – it really just delayed onset of those behaviors). Please be careful in your wording and describe what you observed.

AR: The authors believe that retreat attempts and escape attempts are behavioural indicators of aversion. As the statement that the reviewer is making a reference mentions that “showed a later (p<0.05) onset of behaviours indicative of aversion” we believe that this is not incorrect and we prefer to keep it in the current form. However, acknowledging that aversion may have a broad meaning, we replaced the word aversion by discomfort in the final conclusion of the abstract (L 46).

Section 2.5: Please provide more detail on how you assigned behavior scores. I would assume you would have to watch and score each video repeatedly so that you could observe each animal and it's behaviors (max of 4) on a continuous scale. However, this is my assumption because there is insuffient detail about how each of the 393 and 268 pigs were assessed (or less, since quite a bit of video was lost in the 80N20C group). If this was not the case, what did you do? Scan every x seconds to record what behaviors were done? Document the first time that any of the pigs in the gondola engaged in the behavior? In other words, was the behavioral experimental unit the gondola (115 groups and 76 groups) or the individual animal (393 and 268 pigs)?

AR: We understand this requirement for a more detailed explanation and the following text was included in section 2.: “Behaviour observation was performed individually for each pig, therefore the individual pig being the experimental unit, so each video footage was observed, at least, the number of pigs in the gondola. The same pig was continuously observed throughout the gas exposure until the group of pigs were released.” (L 220).

Line 205: Again, you are measuring potential distress, evidenced by behaviors 2-3 (retreat and struggle), not aversion. Aversion has a specific meaning and requires specific testing to assess. An observational study like this one is not designed to assess diversion. I recommend that you remain consistent with using the behaviors described in Table 2 when discussing behavioral responses.

AR: Escape attempts are considered signs of aversion in pigs. The latest reference making such a statement is as recent as that from Çavuşoğlu, E., Rault, J. L., Gates, R., & Lay, D. C. (2020). Behavioral Response of Weaned Pigs during Gas Euthanasia with CO2, CO2 with Butorphanol, or Nitrous Oxide. Animals, 10(5), 787. We believe that aversion is the behavioural reaction towards noxious or distressing stimluli. Therefore, we prefer to keep indicators of aversion, which can be used to infer negative stimuli such as distress, fear or pain.

Lines 252-253: Please provide the ranges and SE for the measurements taken during the 80N20C stunning. Although the graphs are interesting, knowledge of the actual numbers would be invaluable. Also, what was the spike that occurred at 12:44:00?

AR: The concentrations of the three gases analysed are provided now (L 273). The spike at 12:44 correspond to the beginning of the slaughter period after a lunch break, according to normal functioning of the slaughterhouse. This is now mentioned at the footnote of figure 2.

Table 5: The SEM is reported twice for 80N20C.

AR: SEM duplicates were removed.

Table 6: How many of these had to be restunned? I ask because I would have severe welfare concerns about recommending a method that has a significant risk of requiring restunning, but then ultimately results in a carcass that may not be usable (ultimate waste of animal life).

AR: The number of pigs that had to be re-stunned corresponds to the number of pigs that showed signs of poor stunning. This is now mentioned explicitly in L 334. Carcasses were not harmed due to re-stunning.

Line 349: Instead of “aversive”, I would state that “pigs were not only distressed by exposure to both gas mixes…..”

AR: As already discussed in previous comments we prefer to keep aversion here. Please note that distress is already mentioned in our statement, some words further in the same sentence (L 373).

Line 359: The referenced articles do not appear to be aversion studies (and they are from 1996 and 1977). If you are going to try to make the case that these behaviors correlate to aversion, please reference studies that actually assess these behaviors specific to an aversion response. Preferably from more recent literature, where there has been a lot of work in this area. Otherwise, please remove references to aversion and replace with distress throughout this entire paragraph.

AR: References were changed to Raj 1996 (for being a prominent study on aversion during gas stunning) plus that of Çavuşoğlu et al., 2020 (as a more recent citation).

Lines 404: See comment above for behavioral materials and methods. As there were only a maximum of 4 pigs per gondola, how did you score them without visually tracking each animal individually when doing the scoring? Please be clear on if the experimental unit was the gondola versus the individual animal.

AR: This comment was already addressed in response to comment on section 2.5 (L 220).

Line 417: Brain activity itself does not indicate consciousness. It is well documented that EEG does not correlated to consciousness in humans, so this statement is misleading.

AR: We acknowledge that EEG does not have a lineal correlation with consciousness but instead, provides a level of likeliness of consciousness. According to this, we changed our statement (L 446).

Line 440: In the materials and methods, please describe how you were able to correlate the O2 levels in the chamber to the stunning success of a particular gondola.

AR: In line 265 of the material and methods section already stated that oxygen levels (either above or below 2% in atmospheric air) were kept as covariable in the stun quality model, which gave us the possibility to correlate stun quality with oxygen concentration.

Lines 448-449: I think your results indicate that this needs more investigation, given all of the potential limitations that you have described here for the use of 80N20C. It may show promise, but ethically, I have a hard time believing that this is a welfare refinement if 7% of the animals have to be restunned (welfare issue) and the use of the carcasses are questionable (waste of life) over a well-characterized technique that had no animals that required restunning and no rejection of carcasses.

AR: We agree that a 7% of animals that need to be re-stunned is unaffordable. But, actually, we are anticipating that if the oxygen concentration is kept below 2% it is likely that this percentage would be reduced. We have included a statement that this needs to be investigated further before hypercapnic-hypoxia can be considered as a real alternative to hypercapnia (L 478).

Line 517: How did you define “breathlessness” (and aversion)? There are too many terms coming in during the results and discussions which have not been defined (or measured) in the study design. If you are using them to interpret the data, please describe which behaviors you are using to correlate to these terms. Ideally, you will continue to use the objective terms that you have defined in the materials and methods, instead of subjective interpretations of these behaviors.

AR: Breathlessness is now defined in the introduction section (L 67) as we do not think this is a technical term that should be explained in the Material and Methods but rather a contextualisation of the meaning given to this term.

Reviewer 3 Report

The manuscript titled "Animal welfare and meat quality assessment in gas 2 stunning during commercial slaughter of pigs using 3 80% nitrogen and 20% carbon dioxide compared to 4 90% carbon dioxid” investigate the behaviours that are indicative of pain, fear and discomfort during stunning in a standard group stunning system under commercial conditions with an inhalation of a mixture of 80%N2 and 20%CO2 compared to 90% CO2 and the effects on meat pH and the presence of ecchymosis in the carcass. Effective stunning of animals prior to slaughter is essential to minimize their suffering by preventing the perception of pain and reducing fear. It is certainly an unresolved issue and it is likely to eliminate all the negative aspects of the slaughter of animals. But all efforts made to minimize animal suffering are always welcome. The study is necessary and is correctly planned and well executed and I consider it suitable for publication in the journal. The industry will benefit from its results. I probably miss an economic approach to the study since one of the arguments for not focusing it towards the use of an inert gas such as Argon, which has proven its effectiveness, are economic. I think that making a mention of this aspect of the approach would be welcomed by the users of these promising technologies. I recommend publication with some minor observations detailed in the attached document.

Author Response

Revision Note to manuscript animals-1030363

The comments of the editor and reviewers are given below, with the authors’ response (AR) marked in Italics. Line numbers mentioned by the reviewers refer to the original manuscript, whereas line numbers in the response refer to the revised manuscript unless stated otherwise.

Authors Response (AR): We have amended the manuscript to all suggestions made by reviewers and have detailed the responses to the comments below.

L 100: Referring to the analysis of the quality of the meat is perhaps too broad since since only two quality parameters are analyzed, such as pH and conductivity ... instrumental or sensory quality is not analyzed ... it would probably be more correct to say that these specific parameters are analysed.

AR: This was changed according to reviewer suggestion (L 111).

L 103: Please mention the reference number of the ethics committee report mentioned and the international identification of said committee

AR: The reference number of the ethics comitte was included (L 116).

L106: What sex, age etc. were the pigs? ...

AR: We included the sex of pigs and specified that they were all slaughterweight (L 118). The exacta ge of each pig was not recorded because authors did not have access to records at the farm. Therefore, we prefer not to mention age.

L107: The origin of the pigs is the same and their pre-slaughter management, including transport, was similar?

AR: Acknowledging that the origin of pigs was not the same, we included a statement that pigs came from different farms but the handling procedure was the same for all pigs once at the slaughterhouse (L 121).

L 109: The study analyzes the possibility of stress at slaughter and its consequences.I think it is important to describe other stress factors, so the pre-slaughter handling of slaughtered pigs should be briefly described (Origin, loading, transport, unloading, lairage, fasting time etc. .)

AR: An explanation of the pre-slaughter conditions was included in the material and methods section (L 121).

L 117: Since the economic cost issue is the reason for not being able to use other inert gases such as Argon, it would be very illustrative to give a brief description of the economic cost per animal (or per stunning batch) using the different study gases compared to the best technical option it seems is Argon. In other words, to present a very brief economic analysis of each of the options, including the Argon, which is discarded because it is very expensive.

AR: We included a statement explaining the likely reason of Argon being more expensive than other gases such as carbon dioxide (L 84). However, we prefer not to compare the cost of both gases given that this may change according to providers and markets. We believe that this information will be sufficient for the reader to understand why argon is not used but other inert gases should be used.  

L 171: What indicators were used for this check? ... with what specificity and with what sensitivity? At what times of slaughter procedure was the effectiveness of the stunning checked?

AR: The criteria used to assess stun quality is already described in L 200. The frequency of stun quality checking was included in the text (L 192). Sensitivity and specificity does not apply here as all animals were mionitored by the two assessors at the same time. This is highlighted now in the manuscript (L 193).

L 216: Was the effectiveness of stunning verified at the time of bleeding and during its duration?

AR: Stunning quality monitorisation was perfomed until 60 seconds after sticking provided that no signs of recovery were visible. This is now mentioned in L 194.

L31: bruising was not tested?

AR, No, bruising was not tested because it is not mentioned as a common carcass quality indicator in the literature about product quality and gas stunning.

L 429: how many were dead after stun? ...

AR: Whether pigs were dead or alive could not be determined in this study. Cessation of brain activity or hear frequency are the gold Standard indicators to confirm death. However, neither brain nor heart activity could be monitored. Therefore, despite we hypothesise that pigs exposed to hypercapnia were all dead we prefer not to explícit it as it is just a hypothesies that would need confirmation.

L 527: "The presence of ecchymosis was used to evaluate carcass quality"... I think that in the conclusions it cannot be said that it does not affect the quality of the carcass, they should refer only to ecchymoses since the quality of the carcass is much more than the presence or not of ecchymosis.

AR: We agree with this comment, so we decided to change the conclusions according to reviewer’s concern (L 564).

L 530: In practice, do the authors recommend the use of this methodology, including the economic aspect of the proposed technology in making decisions? ...

AR: Authors prefer not to recommend or dissuade the use of this alternative gas, but to limit our words into the description of the results achieved within this trial. We believe that decision should be made according to numerous criteria, including aversion, stun quality, product quality parameters, but it is up to stakeholders whether results are suitable enough to drive a change or not.

L 530: From an economic point of view, which of the tested technologies is the most recommended by authors?

AR: Our study did not include a cost-benefit analysis. Therefore we do not feel qualified to make such a judgement.